# Modeling Uncertainty with Hedged Instance Embedding

**Seong Joon Oh**[*]
LINE Corporation
coallaoh@linecorp.com

**Kevin Murphy**
Google Research
kpmurphy@google.com

**Jiyan Pan**
Google Research
jiyanpan@google.com

**Joseph Roth**
Google Research
josephroth@google.com

**Florian Schroff**
Google Research
fschroff@google.com

**Andrew Gallagher**
Google Research
agallagher@google.com

## Abstract

Instance embeddings are an efficient and versatile image representation that facilitates applications like recognition, verification, retrieval, and clustering. Many metric learning methods represent the input as a single point in the embedding space. Often the distance between points is used as a proxy for match confidence. However, this can fail to represent uncertainty which can arise when the input is ambiguous, e.g., due to occlusion or blurriness. This work addresses this issue and explicitly models the uncertainty by "hedging" the location of each input in the embedding space. We introduce the *hedged instance embedding* (*HIB*) in which embeddings are modeled as random variables and the model is trained under the variational information bottleneck principle (Alemi et al., 2016; Achille & Soatto, 2018). Empirical results on our new *N-digit MNIST* dataset show that our method leads to the desired behavior of "hedging its bets" across the embedding space upon encountering ambiguous inputs. This results in improved performance for image matching and classification tasks, more structure in the learned embedding space, and an ability to compute a per-exemplar uncertainty measure which is correlated with downstream performance.

## 1 Introduction

An instance embedding is a mapping $f$ from an input $x$, such as an image, to a vector representation, $z \in \mathbb{R}^D$, such that "similar" inputs are mapped to nearby points in space. Embeddings are a versatile representation that support various downstream tasks, including image retrieval (Babenko et al., 2014) and face recognition (Schroff et al., 2015).

Instance embeddings are often treated deterministically, *i.e.*, $z = f(x)$ is a point in $\mathbb{R}^D$. We refer to this approach as a *point embedding*. One drawback of this representation is the difficulty of modeling aleatoric uncertainty (Kendall & Gal, 2017), *i.e.* uncertainty induced by the input. In the case of images this can be caused by occlusion, blurriness, low-contrast and other factors.

To illustrate this, consider the example in Figure 1a. On the left, we show an image composed of two adjacent MNIST digits, the first of which is highly occluded. The right digit is clearly a 7, but the left digit could be a 1, or a 4. One way to express this uncertainty about which choice to make is to map the input to a region of space, representing the inherent uncertainty of "where it belongs".

We propose a new method, called *hedged instance embedding* (*HIB*), which achieves this goal. Each embedding is represented as a random variable, $Z \sim p(z|x) \in \mathbb{R}^D$. The embedding effectively spreads probability mass across locations in space, depending on the level of uncertainty. For example in Figure 1b, the corrupted image is mapped to a two-component mixture of Gaussians covering both the "17" and "47" clusters. We propose a training scheme for the HIB with a learnable-margin contrastive loss and the variational information bottleneck (VIB) principle (Alemi et al., 2016; Achille & Soatto, 2018).

---

[*]Work performed during an internship with Google Research.

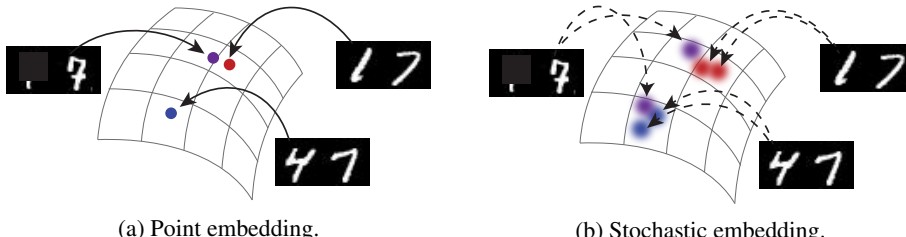

(a) Point embedding.       (b) Stochastic embedding.

Figure 1: Unlike point embeddings, stochastic embeddings may hedge their bets across the space. When both "17" and "47" are plausible, our 2-component Gaussian mixture embedding has the power to spread probability mass on clusters with clean "17" and "47" images. By contrast, the point embedding will choose to be close to one or the other of these points (or somewhere between).

To evaluate our method, we propose a novel dataset, N-digit MNIST, which we will open source. Using this dataset, we show that HIB exhibits several desirable properties compared to point embeddings: (1) downstream task performance (*e.g.* recognition and verification) improves for uncertain inputs; (2) the embedding space exhibits enhanced structural regularity; and (3) a per-exemplar uncertainty measure that predicts when the output of the system is reliable.

## 2 METHODS

In this section, we describe our method in detail.

### 2.1 POINT EMBEDDINGS

Standard point embedding methods try to compute embeddings such that $z_1 = f(x_1)$ and $z_2 = f(x_2)$ are "close" in the embedding space if $x_1$ and $x_2$ are "similar" in the ambient space. To obtain such a mapping, we must decide on the definition of "closeness" as well as a training objective, as we explain below.

**Contrastive loss** Contrastive loss (Hadsell et al., 2006) is designed to encourage a small Euclidean distance between a similar pair, and large distance of margin $M > 0$ for a dissimilar pair. The loss is

$$\mathcal{L}_{\mathrm{con}} = \begin{cases} ||z_1 - z_2||_2^2 & \text{if match} \\ \max(M - ||z_1 - z_2||_2, 0)^2 & \text{if non-match} \end{cases} \tag{1}$$

where $z_i = f(x_i)$. The hyperparameter $M$ is usually set heuristically or based on validation-set performance.

**Soft contrastive loss** A probabilistic alternative to contrastive loss, which we will use in our experiments is defined here. It represents the probability that a pair of points is matching:

$$p(m|z_1, z_2) := \sigma(-a||z_1 - z_2||_2 + b) \tag{2}$$

with scalar parameters $a > 0$ and $b \in \mathbb{R}$, and the sigmoid function $\sigma(t) = \frac{1}{1+e^{-t}}$. This formulation calibrates Euclidean distances into a probabilistic expression for similarity. Instead of setting a hard threshold like $M$, $a$ and $b$ together comprise a soft threshold on the Euclidean distance. We will later let $a$ and $b$ be trained from data.

Having defined the match probability $p(m|z_1, z_2)$, we formulate the contrastive loss as a binary classification loss based on the softmax cross-entropy (negative log-likelihood loss). More precisely, for an embedding pair $(z_1, z_2)$ the loss is defined as

$$\mathcal{L}_{\mathrm{softcon}} = -\log p(m = \hat{m}|z_1, z_2) = \begin{cases} -\log p(m|z_1, z_2) & \text{if } \hat{m} = 1, \\ -\log (1 - p(m|z_1, z_2)) & \text{if } \hat{m} = 0, \end{cases} \tag{3}$$

where $\hat{m}$ is the indicator function with value 1 for ground-truth match and 0 otherwise.

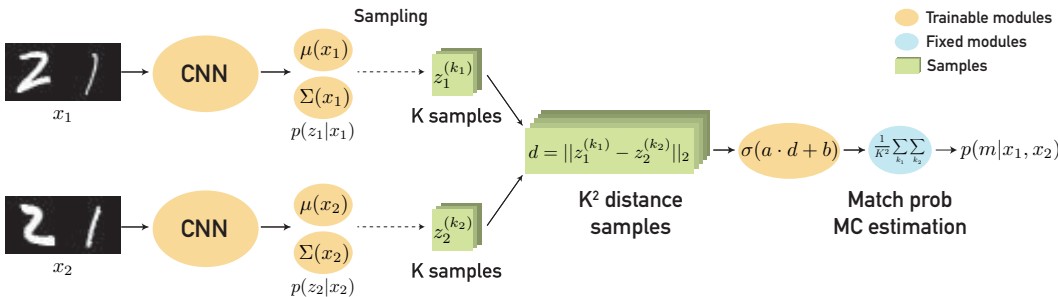

Figure 2: Computational graph for computing $p(m|x_1, x_2)$ using HIB with Gaussian embeddings.

Although some prior work has explored this soft contrastive loss (*e.g.* Bertinetto et al. (2016); Orekondy et al. (2018)), it does not seem to be widely used. However, in our experiments, it performs strictly better than the hard margin version, as explained in Appendix B.

## 2.2 STOCHASTIC EMBEDDINGS

In HIB, we treat embeddings as stochastic mappings $x \mapsto Z$, and write the distribution as $Z \sim p(z|x)$. In the sections below, we show how to learn and use this mapping.

**Match probability for probabilistic embeddings** The probability of two inputs matching, given in Equation 2, can easily be extended to stochastic embeddings, as follows:

$$p(m|x_1, x_2) = \int p(m|z_1, z_2)p(z_1|x_1)p(z_2|x_2)\, \mathrm{d}z_1 \mathrm{d}z_2. \tag{4}$$

We approximate this integral via Monte-Carlo sampling from $z_1^{(k_1)} \sim p(z_1|x_1)$ and $z_2^{(k_2)} \sim p(z_2|x_2)$:

$$p(m|x_1, x_2) \approx \frac{1}{K^2} \sum_{k_1=1}^{K} \sum_{k_2=1}^{K} p\left(m|z_1^{(k_1)}, z_2^{(k_2)}\right). \tag{5}$$

In practice, we get good results using $K = 8$ samples per input image. Now we discuss the computation of $p(z|x)$.

**Single Gaussian embedding** The simplest setting is to let $p(z|x)$ be a $D$-dimensional Gaussian with mean $\mu(x)$ and diagonal covariance $\Sigma(x)$, where $\mu$ and $\Sigma$ are computed via a deep neural network with a shared "body" and $2D$ total outputs. Given a Gaussian representation, we can draw $K$ samples $z^{(1)}, \cdots, z^{(K)} \overset{\text{iid}}{\sim} p(z|x)$, which we can use to approximate the match probability. Furthermore, we can use the reparametrization trick (Kingma & Welling, 2013) to rewrite the samples as $z^{(k)} = \text{diag}\left(\sqrt{\Sigma(x)}\right) \cdot \epsilon^{(k)} + \mu(x)$, where $\epsilon^{(1)}, \cdots, \epsilon^{(K)} \overset{\text{iid}}{\sim} N(0, I)$. This enables easy backpropagation during training.

**Mixture of Gaussians (MoG) embedding** We can obtain a more flexible representation of uncertainty by using a mixture of $C$ Gaussians to represent our embeddings, *i.e.* $p(z|x) = \sum_{c=1}^{C} \mathcal{N}(z; \mu(x,c), \Sigma(x,c))$. To enhance computational efficiency, the $2C$ mappings $\{(\mu(x,c), \Sigma(x,c))\}_{c=1}^{C}$ share a common CNN stump and are branched with one linear layer per branch. When approximating Equation 5, we use stratified sampling, *i.e.* we sample the same number of samples from each Gaussian component.

**Computational considerations** The overall pipeline for computing the match probability is shown in Figure 2. If we use a single Gaussian embedding, the cost (time complexity) of computing the stochastic representation is essentially the same as for point embedding methods, due to the use of a shared network for computing $\mu(x)$ and $\Sigma(x)$. Also, the space requirement is only $2\times$ more. (This is an important consideration for many embedding-based methods.)

## 2.3 VIB TRAINING OBJECTIVE

For training our stochastic embedding, we combine two ingredients: soft contrastive loss in Equation 3 and the VIB principle Alemi et al. (2016); Achille & Soatto (2018). We start with a summary of the original VIB formulation, and then describe its extension to our setting.

**Variational Information Bottleneck (VIB)**   A discriminative model $p(y|x)$ is trained under the information bottleneck principle (Tishby et al., 1999) by maximizing the following objective:

$$I(z, y) - \beta I(z, x) \tag{6}$$

where $I$ is the mutual information, and $\beta > 0$ is a hyperparameter which controls the tradeoff between the sufficiency of $z$ for predicting $y$, and the minimality (size) of the representation. Intuitively, this objective lets the latent encoding $z$ capture the salient parts of $x$ (salient for predicting $y$), while disallowing it to "memorise" other parts of the input which are irrelevant.

Computing the mutual information is generally computationally intractable, but it is possible to use a tractable variational approximation as shown in Alemi et al. (2016); Achille & Soatto (2018). In particular, under the Markov assumption that $p(z|x, y) = p(z|x)$ we arrive at a lower bound on Equation 6 for every training data point $(x, y)$ as follows:

$$-\mathcal{L}_{\text{VIB}} := \mathbb{E}_{z \sim p(z|x)} \left[ \log q(y|z) \right] - \beta \cdot \text{KL}(p(z|x) \| r(z)) \tag{7}$$

where $p(z|x)$ is the latent distribution for $x$, $q(y|z)$ is the decoder (classifier), and $r(z)$ is an approximate marginal term that is typically set to the unit Gaussian $\mathcal{N}(0, I)$.

In Alemi et al. (2016), this approach was shown (experimentally) to be more robust to adversarial image perturbations than deterministic classifiers. It has also been shown to provide a useful way to detect out-of-domain inputs (Alemi et al., 2018). Hence we use it as the foundation for our approach.

**VIB for learning stochastic embeddings**   We now apply the above method to learn our stochastic embedding. In particular, we train a discriminative model based on matching or mismatching pairs of inputs $(x_1, x_2)$, by minimizing the following loss:

$$\mathcal{L}_{\text{VIBEmb}} := - \mathbb{E}_{z_1 \sim p(z_1|x_1), z_2 \sim p(z_2|x_2)} \left[ \log p(m = \hat{m}|z_1, z_2) \right]$$
$$+ \beta \cdot \left[ \text{KL}(p(z_1|x_1) \| r(z_1)) + \text{KL}(p(z_2|x_2) \| r(z_2)) \right] \tag{8}$$

where the first term is given by the negative log likelihood loss with respect to the ground truth match $\hat{m}$ (this is identical to Equation 3, the soft contrastive loss), and the second term is the KL regularization term, $r(z) = \mathcal{N}(z; 0, I)$. The full derivation is in appendix F. We optimize this loss with respect to the embedding function $(\mu(x), \Sigma(x))$, as well as with respect to the $a$ and $b$ terms in the match probability in Equation 2.

Note that most pairs are not matching, so the $m = 1$ class is rare. To handle this, we encourage a balance of $m = 0$ and $m = 1$ pair samples within each SGD minibatch by using two streams of input sample images. One samples images from the training set at random and the other selects images from specific class labels, and then these are randomly shuffled to produce the final batch. As a result, each minibatch has plenty of positive pairs even when there are a large number of classes.

## 2.4 UNCERTAINTY MEASURE

One useful property of our method is that the embedding is a distribution and encodes the level of uncertainty for given inputs. As a scalar uncertainty measure, we propose the *self-mismatch* probability as follows:

$$\eta(x) := 1 - p(m|x, x) \geq 0 \tag{9}$$

Intuitively, the embedding for an ambiguous input will span diverse semantic classes (as in Figure 1b). $\eta(x)$ quantifies this by measuring the chance two samples of the embedding $z_1, z_2 \overset{\text{iid}}{\sim} p(z|x)$ belong to different semantic classes (i.e., the event $m = 0$ happens). We compute $\eta(x)$ using the Monte-Carlo estimation in Equation 5.

Prior works (Vilnis & McCallum, 2014; Bojchevski & Günnemann, 2018) have computed uncertainty for Gaussian embeddings based on volumetric measures like trace or determinant of covariance matrix. Unlike those measures, $\eta(x)$ can be computed for *any* distribution from which one can sample, including multi-modal distributions like mixture of Gaussians.

## 3 RELATED WORK

In this section, we mention the most closely related work from the fields of deep learning and probabilistic modeling.

**Probabilistic DNNs** Several works have considered the problem of estimating the uncertainty of a regression or classification model, $p(y|x)$, when given ambiguous inputs. One of the simplest and most widely used techniques is known as Monte Carlo dropout (Gal & Ghahramani, 2016). In this approach, different random components of the hidden activations are "masked out" and a distribution over the outputs $f(x)$ is computed. By contrast, we compute a parametric representation of the uncertainty and use Monte Carlo to approximate the probability of two points matching. Monte Carlo dropout is not directly applicable in our setting as the randomness is attached to model parameters and is independent of input; it is designed to measure model uncertainty (epistemic uncertainty). On the other hand, we measure input uncertainty where the embedding distribution is conditioned on the input. Our model is designed to measure input uncertainty (aleatoric uncertainty).

**VAEs and VIB** A variational autoencoder (VAE, Kingma & Welling (2013)) is a latent variable model of the form $p(x, z) = p(z)p(x|z)$, in which the generative decoder $p(x|z)$ and an encoder network, $q(z|x)$ are trained jointly so as to maximize the evidence lower bound. By contrast, we compute a discriminative model on pairs of inputs to maximize a lower bound on the match probability. The variational information bottleneck (VIB) method (Alemi et al., 2016; Achille & Soatto, 2018) uses a variational approximation similar to the VAE to approximate the information bottleneck objective (Tishby et al., 1999). We build on this as explained in 2.3.

**Point embeddings** Instance embeddings are often trained with metric learning objectives, such as contrastive (Hadsell et al., 2006) and triplet (Schroff et al., 2015) losses. Although these methods work well, they require careful sampling schemes (Wu et al., 2017; Movshovitz-Attias et al., 2017). Many other alternatives have attempted to decouple the dependency on sampling, including softmax cross-entropy loss coupled with the centre loss (Wan et al., 2018), or a clustering-based loss (Song et al., 2017), and have improved the embedding quality. In HIB, we use a soft contrastive loss, as explained in section 2.1.

**Probabilistic embeddings** The idea of probabilistic embeddings is not new. For example, Vilnis & McCallum (2014) proposed Gaussian embeddings to represent levels of specificity of word embeddings (*e.g.* "Bach" is more specific than "composer"). The closeness of the two Gaussians is based on their KL-divergence, and uncertainty is computed from the spread of Gaussian (determinant of covariance matrix). See also Karaletsos et al. (2015); Bojchevski & Günnemann (2018) for related work. Neelakantan et al. (2014) proposed to represent each word using multiple prototypes, using a "best of $K$" loss when training. HIB, on the other hand, measures closeness based on a quantity related to the expected Euclidean distance, and measures uncertainty using the *self-mismatch* probability.

## 4 EXPERIMENTS

In this section, we report our experimental results, where we compare our stochastic embeddings to point embeddings. We consider two main tasks: the verification task (*i.e.*, determining if two input images correspond to the same class or not), and the identification task (*i.e.*, predicting the label for an input image). For the latter, we use a K-nearest neighbors approach with $K = 5$. We compare performance of three methods: a baseline deterministic embedding method, our stochastic embedding method with a Gaussian embedding, and our stochastic embedding method with a mixture of Gaussians embedding. We also conduct a qualitative comparison of the embeddings of each method.

In the Appendix, Section C, we describe additional experiments, including where HIB is applied to a dataset of cat and dog images of over 100K distinct animals and the embedding is directed towards identifying specific animals.

### 4.1 EXPERIMENTAL DETAILS

We conduct all our experiments on a new dataset we created called N-digit MNIST, which consists of images composed of $N$ adjacent MNIST digits, which may be randomly occluded (partially or fully). See appendix A for details. During training, we occlude 20% of the digits independently. A single image can have multiple corrupted digits. During testing, we consider both clean (unoccluded) and corrupted (occluded) images, and report results separately. We use images with $N = 2$ and $N = 3$ digits. We will open source the data to ensure reproducibility.

Since our dataset is fairly simple, we use a shallow CNN model to compute the embedding function. Specifically, it consists of 2 convolutional layers, with $5 \times 5$ filters, each followed by max pooling, and then a fully connected layer mapping to $D$ dimensions. We focus on the cases where $D = 2$ or $D = 3$, and present additional results where $D = 4$ in the Appendix. When we use more dimensions, we find that all methods (both stochastic and deterministic) perform almost perfectly (upper 90%), so there are no interesting differences to report. We leave exploration of more challenging datasets, and higher dimensional embeddings, to future work.

Our networks are built with TensorFlow (Abadi et al., 2015). For each task, the input is an image of size $28 \times (N \times 28)$, where N is the number of concatenated digit images. We use a batch size of 128 and 500k training iterations. Each model is trained from scratch with random weight initialization. The KL-divergence hyperparameter $\beta$ is set to $10^{-4}$ throughout the experiments. We report effects of changing $\beta$ in appendix E.

### 4.2 QUALITATIVE EVALUATION OF THE REPRESENTATION

Figure 3 shows HIB 2D Gaussian embeddings for the clean and corrupt subsets of the test set. We can easily see that the corrupt images generally have larger (*i.e.*, less certain) embeddings. In the Appendix, Figure 7 shows a similar result when using a 2D MoG representation, and Figure 8 shows a similar result for 3D Gaussian embeddings.

Figure 4 illustrates the embeddings for several test set images, overlaid with an indication of each class' centroid. Hedged embeddings capture the uncertainty that may exist across complex subsets of the class label space, by learning a layout of the embedding space such that classes that may be confused are able to receive density from the underlying hedged embedding distribution.

We observe enhanced spatial regularity when using HIB. Classes with a common least or most significant digit roughly align parallel to the $x$ or $y$ axis. This is because of the diagonal structure of the embedding covariance matrix. By controlling the parametrization of the covariance matrix, one may apply varying degrees and types of structures over the embedding space (*e.g.* diagonally aligned embeddings). See appendix D for more analysis of the learned latent space.

### 4.3 QUANTITATIVE EVALUATION OF THE BENEFITS OF STOCHASTIC EMBEDDING

We first measure performance on the verification task, where the network is used to compute $p(m|x_1, x_2)$ for 10k pairs of test images, half of which are matches and half are not. The average precision (AP) of this task is reported in the top half of Table 1. HIB shows improved performance, especially for corrupted test images. For example, in the $N = 2$ digit case, when using $D = 2$ dimensions, point embeddings achieve 88.0% AP on corrupt test images, while hedged instance embeddings improves to 90.7% with $C = 1$ Gaussian, and 91.2% with $C = 2$ Gaussians.

We next measure performance on the KNN identification task. The bottom half of Table 1 reports the results. Again, proposed stochastic embeddings generally outperform point embeddings, with the greatest advantage for the corrupted input samples. For example, in the $N = 2$ digit case, when using $D = 2$ dimensions, point embeddings achieve 58.3% AP on corrupt test images, while HIB improves to 76.0% with $C = 1$ Gaussian, and 75.7% with $C = 2$ Gaussians.

### 4.4 KNOWN UNKNOWNS

In this section, we address the task of estimating when an input can be reliably recognized or not, which has important practical applications. To do this, we use the measure of uncertainty $\eta(x)$ defined in Equation 9.

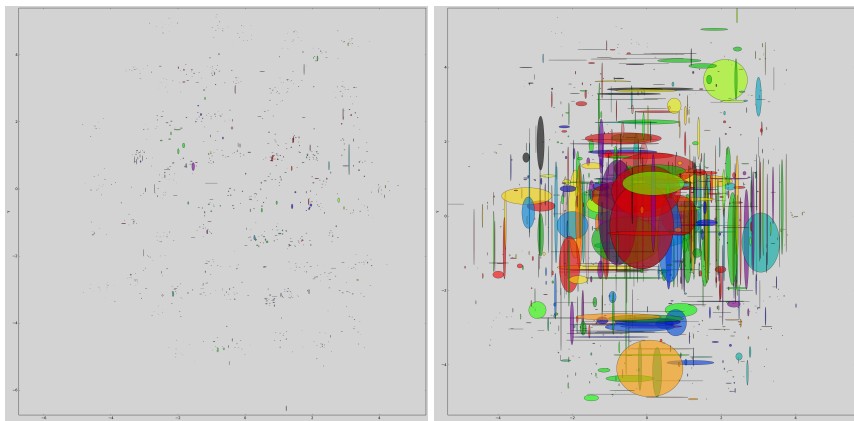

Figure 3: 2D Gaussian embeddings from 2-digit MNIST. Corrupted test images (right) generally have embeddings that spread density across larger regions of the space than clean images (left). See Appendix Figure 8 for a 3D version.

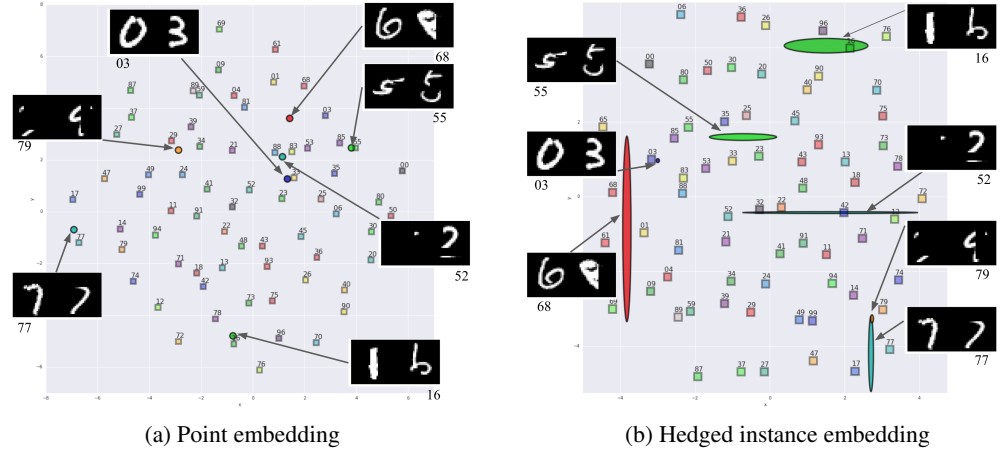

(a) Point embedding                                    (b) Hedged instance embedding

Figure 4: Point vs. hedged image embeddings. Square markers indicate the class centroids of embeddings for 2-digit MNIST. **Left:** Centroids learned with point embeddings. Occluded examples map into the embedding space with no special indication of corruption or confidence. **Right:** Hedged embeddings comprising a single Gaussian (rendered as $3\sigma$ ellipses). A clean input such as the "03" is embedded into a tight Gaussian, while a corrupted image such as the "52" occupies an elongated ellipse that places bets that the least significant digit is a "2" but the most significant digit could be any of a number of choices.

We measure the utility of $\eta(x)$ for the identification task as follows. For the test set, we sort all test input examples according to $\eta(x)$, and bin examples into 20 bins ranging from the lowest to highest range of uncertainty. We then measure the KNN classification accuracy for the examples falling in each bin.

To measure the utility of $\eta(x)$ for the verification task, we take random pairs of samples, $(x_1, x_2)$, and compute the mean of their uncertainties, $\eta(x_1, x_2) = \frac{1}{2}(\eta(x_1) + \eta(x_2))$. We then distribute the test pairs to 20 equal-sized bins according to their uncertainty levels, and compute the probability of a match for each pair. To cope with the severe class imbalance (most pairs don't match), we measure performance for each bin using average precision (AP). Then, again, the Kendall's tau is applied to measure the uncertainty-performance correlation.

| | $N = 2, D = 2$ | | | $N = 2, D = 3$ | | | $N = 3, D = 2$ | | | $N = 3, D = 3$ | | |
|---|---|---|---|---|---|---|---|---|---|---|---|---|
| | point | MoG-1 | MoG-2 | point | MoG-1 | MoG-2 | point | MoG-1 | MoG-2 | point | MoG-1 | MoG-2 |
| **Verification AP** | | | | | | | | | | | | |
| clean | 0.987 | 0.989 | 0.990 | 0.996 | 0.996 | 0.996 | 0.978 | 0.981 | 0.976 | 0.987 | 0.989 | 0.991 |
| corrupt | 0.880 | 0.907 | 0.912 | 0.913 | 0.926 | 0.932 | 0.886 | 0.899 | 0.904 | 0.901 | 0.922 | 0.925 |
| **KNN Accuracy** | | | | | | | | | | | | |
| clean | 0.871 | 0.879 | 0.888 | 0.942 | 0.953 | 0.939 | 0.554 | 0.591 | 0.540 | 0.795 | 0.770 | 0.766 |
| corrupt | 0.583 | 0.760 | 0.757 | 0.874 | 0.909 | 0.885 | 0.274 | 0.350 | 0.351 | 0.522 | 0.555 | 0.598 |

Table 1: Accuracy of pairwise verification and KNN identification tasks for point embeddings, and our hedged embeddings with a single Gaussian component (MoG-1) and two components (MoG-2). We report results for images with $N$ digits and using $D$ embedding dimensions.

| | $N = 2, D = 2$ | | $N = 2, D = 3$ | | $N = 3, D = 2$ | | $N = 3, D = 3$ | |
|---|---|---|---|---|---|---|---|---|
| | MoG-1 | MoG-2 | MoG-1 | MoG-2 | MoG-1 | MoG-2 | MoG-1 | MoG-2 |
| **AP Correlation** | | | | | | | | |
| clean | 0.74 | 0.43 | 0.68 | 0.48 | 0.63 | 0.28 | 0.51 | 0.39 |
| corrupt | 0.81 | 0.79 | 0.86 | 0.87 | 0.82 | 0.76 | 0.85 | 0.79 |
| **KNN Correlation** | | | | | | | | |
| clean | 0.71 | 0.57 | 0.72 | 0.47 | 0.76 | 0.29 | 0.74 | 0.54 |
| corrupt | 0.47 | 0.43 | 0.55 | 0.52 | 0.49 | 0.50 | 0.67 | 0.34 |

Table 2: Correlations between each input image's measure of uncertainty, $\eta(x)$, and AP and KNN performances. High correlation coefficients suggest a close relationship.

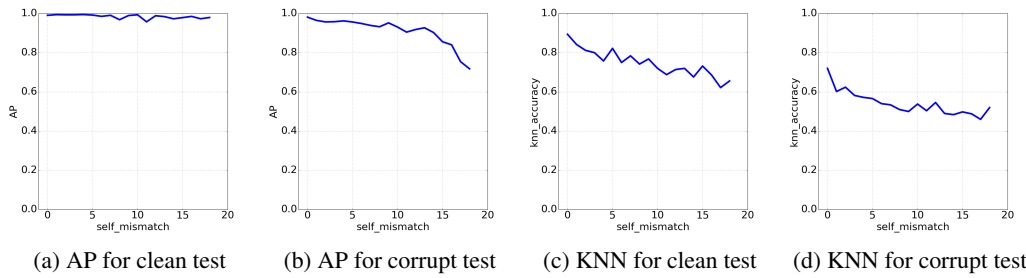

(a) AP for clean test     (b) AP for corrupt test     (c) KNN for clean test     (d) KNN for corrupt test

Figure 5: Correlations between the uncertainty measure $\eta(x)$ and AP and KNN accuracy on the test set for the $N = 3$, $D = 3$ case using single Gaussian embeddings. Uncertainty increases along the horizontal axis. We observe that accuracy generally decreases as uncertainty increases.

Figure 5 plots the AP and KNN accuracy vs the uncertainty bin index, for both clean and corrupted inputs. We see that when the performance drops off, the model's uncertainty measure increases, as desired.

To quantify this, we compute the correlation between the performance metric and the uncertainty metric. Instead of the standard linear correlations (Pearson correlation coefficient), we use Kendall's tau correlation (Kendall, 1938) that measures the degree of monotonicity between the performance and the uncertainty level (bin index), inverting the sign so that positive correlation aligns with our goal. The results of different models are shown in Table 2. In general, the measure $\eta(x)$ correlates with the task performance. As a baseline for point embeddings in KNN, we explored using the distance to the nearest neighbor as a proxy for uncertainty, but found that it performed poorly. The HIB uncertainty metric correlates with task accuracy even in within the subset of clean (uncorrupted) input images having no corrupted digits, indicating that HIB's understanding of uncertainty goes beyond simply detecting which images are corrupted.

## 5 DISCUSSION AND FUTURE WORK

Hedged instance embedding is a stochastic embedding that captures the uncertainty of the mapping of an image to a latent embedding space, by spreading density across plausible locations. This

results in improved performance on various tasks, such as verification and identification, especially for ambiguous corrupted input. It also allows for a simple way to estimate the uncertainty of the embedding that is correlated with performance on downstream tasks.

There are many possible directions for future work, including experimenting with higher-dimensional embeddings, and harder datasets. As an early look at these tasks, in the Appendix, Section C.3, we apply HIB towards cat and dog instance embedding directed towards identifying specific animals with 20D embeddings. It would also be interesting to consider the "open world" (or "unknown unknowns") scenario, in which the test set may contain examples of novel classes, such as digit combinations that were not in the training set (see e.g., Lakkaraju et al. (2017); Günther et al. (2017)). This is likely to result in uncertainty about where to embed the input which is different from the uncertainty induced by occlusion, since uncertainty due to open world is *epistemic* (due to lack of knowledge of a class), whereas uncertainty due to occlusion is *aleatoric* (intrinsic, due to lack of information in the input), as explained in Kendall & Gal (2017). Preliminary experiments suggest that $\eta(x)$ correlates well with detecting occluded inputs, but does not work as well for novel classes. We leave more detailed modeling of epistemic uncertainty as future work.

### ACKNOWLEDGMENTS

We are grateful to Alex Alemi and Josh Dillon, helpful with discussions related to VAEs and Variational Information Bottleneck, and to Lucy Gao for the pet dataset of cats and dogs.

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

# Appendix

## A  N-DIGIT MNIST DATASET

We present N-digit MNIST, a new dataset based upon MNIST (LeCun, 1998) that has an exponentially large number of classes on the number of digits $N$, for which embedding-style classification methods are well-suited. The dataset is created by horizontally concatenating $N$ MNIST digit images. While constructing new classes, we respect the training and test splits. For example, a test image from 2-digit MNIST of a "54" does not have its "5" or its "4" shared with any image from the training set (in all positions).

| Number Digits | Total Classes | Training Classes | Unseen Test Classes | Seen Test Classes | Training Images | Test Images |
|---|---|---|---|---|---|---|
| 2 | 100 | 70 | 30 | 70 | 100 000 | 10 000 |
| 3 | 1000 | 700 | 300 | 700 | 100 000 | 10 000 |

Table 3: Summary of N-digit MNIST dataset for $N = 2, 3$.

We employ 2- and 3-digit MNIST (Table 3) in our experiments. This dataset is meant to provide a test bed for easier and more efficient evaluation of embedding algorithms than with larger and more realistic datasets. N-digit MNIST is more suitable for embedding evaluation than other synthetic datasets due to the exponentially increasing number of classes as well as the factorizability aspect: each digit position corresponds to *e.g.* a face attribute for face datasets.

We inject uncertainty into the underlying tasks by randomly occluding (with black fill-in) regions of images at training and test time. Specifically, the corruption operation is done independently on each digit of number samples in the dataset. A random-sized square patch is identified from a random location of each $28 \times 28$ digit image. The patch side length is first sampled $L \sim \text{Unif}(0, 28)$, and then the top left patch corner coordinates are sampled $(TL_x, TL_y) \overset{\text{iid}}{\sim} \text{Unif}(0, 28 - L)$, so that the occluded square size is always $L^2$. During training, we set independent binary random flags $B$ for every digit determining whether to perform the occlusion at all; the occlusion chance is set to 20%. For testing, we prepare twin datasets, clean and corrupt, with digit images that are either not corrupted with occlusion at all or always occluded, respectively.

## B  SOFT CONTRASTIVE LOSS VERSUS CONTRASTIVE LOSS

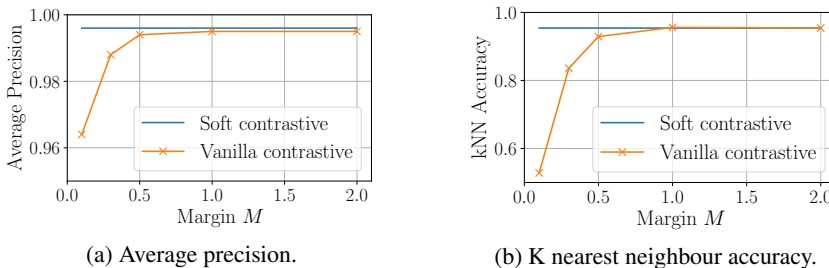

(a) Average precision.

(b) K nearest neighbour accuracy.

Figure 6: Soft contrastive versus vanilla contrastive loss for point embeddings.

As a building block for the HIB, soft contrastive loss has been proposed in §2.1. Soft contrastive loss has a conceptual advantage over the vanilla contrastive loss that the margin hyperparameter $M$ does not have to be hand-tuned. Here we verify that soft contrastive loss outperforms the vanilla version over a range of $M$ values.

Figure 9 shows the verification (average precision) and identification (KNN accuracy) performance of embedding 2-digit MNIST samples. In both evaluations, soft contrastive loss performance is upper bounding the vanilla contrastive case. This new formulation removes one hyperparameter from the learning process, while not sacrificing performance.

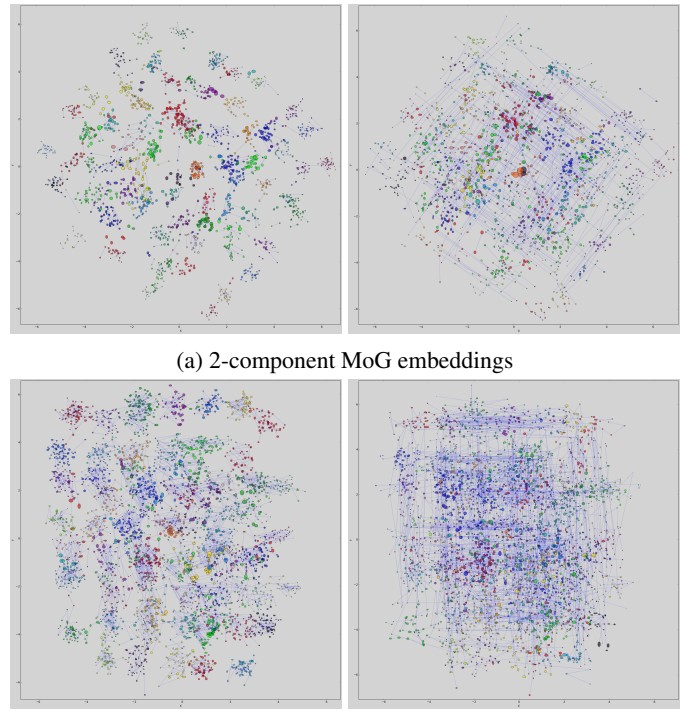

(a) 2-component MoG embeddings

(b) 4-component MoG embeddings

Figure 7: MoG embeddings of 2-digit numbers. Components corresponding to common input are joined with a thin line. It is qualitatively apparent that corrupted examples (right) tend to have more dispersed components than clean ones (left) as a result of the "bet hedging" behavior.

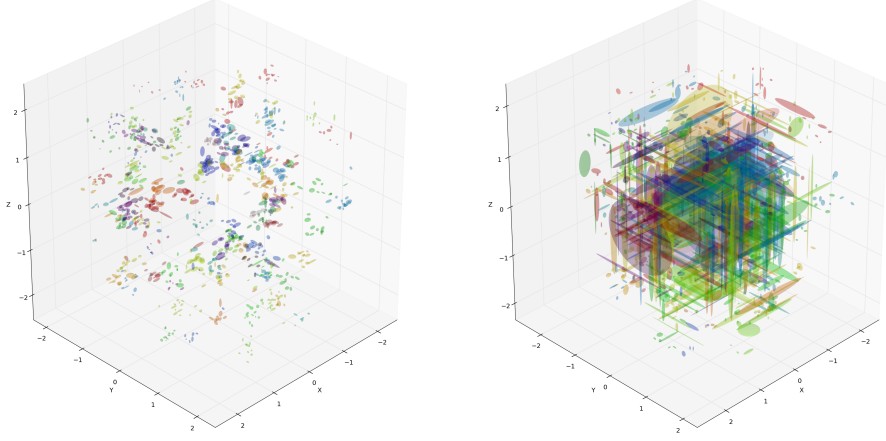

Figure 8: 3D embeddings of 3-digit MNIST. Corrupted test images (right) generally have embeddings that spread density across larger regions of the space than those of clean images (left).

## C  ADDITIONAL RESULTS

In this section, we include some extra results which would not fit in the main paper.

|  | $N = 3, D = 4$ | | |
| --- | --- | --- | --- |
|  | point | MoG-1 | MoG-2 |
| **Verification AP** | | | |
| clean | 0.996 | 0.996 | 0.995 |
| corrupt | 0.942 | 0.944 | 0.948 |
| **KNN Accuracy** | | | |
| clean | 0.914 | 0.917 | 0.922 |
| corrupt | 0.803 | 0.816 | 0.809 |

|  | $N = 3, D = 4$ | |
| --- | --- | --- |
|  | MoG-1 | MoG-2 |
| **AP Correlation** | | |
| clean | 0.35 | 0.33 |
| corrupt | 0.79 | 0.68 |
| **KNN Correlation** | | |
| clean | 0.31 | 0.32 |
| corrupt | 0.42 | 0.35 |

(a) Task accuracies for pairwise verification and KNN identification.

(b) Correlations between uncertainty, $\eta(x)$, and task performances.

Table 4: Task performance and uncertainty measure correlations with task performance for 4D embeddings with 3-digit MNIST.

## C.1 VISUAL REPRESENTATIONS OF MNIST MoG EMBEDDINGS

Figure 7 shows some 2D embeddings of 2-digit images using a MoG representation, with $C = 2$ or $C = 4$ clusters per embedding. Figure 8 shows some 3D embeddings of 3-digit images using a single Gaussian.

## C.2 4D MNIST EMBEDDINGS

In Table 4, we report the results of 4D embeddings with 3-digit MNIST. Here, the task performance begins to saturate, with verification accuracy on clean input images above 0.99. However, we again observe that HIB provides a slight performance improvement over point embeddings for corrupt images, and task performance correlates with uncertainty.

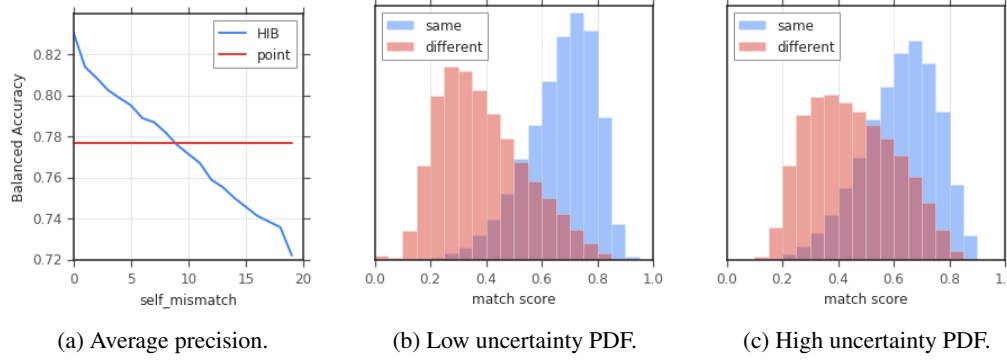

(a) Average precision.  (b) Low uncertainty PDF.  (c) High uncertainty PDF.

Figure 9: Correlation between the uncertainty measure $\eta(x)$ and balanced accuracy on the pet test set for 20D single Gaussian embeddings. Uncertainty increases along the horizontal axis. (b) and (c) show match score distributions of pairs of the same and different pets for lowest and highest uncertainty bins. There is clearly more confusion for the highly uncertain samples.

## C.3 HIB EMBEDDINGS FOR PET IDENTIFICATION

We applied HIB to learn instance embeddings for identifying pets, using an internal dataset of nearly 1M cat and dog images with known identity, including 117913 different pets. We trained an HIB with 20D embeddings and a single Gaussian component (diagonal covariance). The CNN portion of the model is a MobileNet (Howard et al., 2017) with a width multiplier of 0.25. No artificial corruption is applied to the images, as there is sufficient uncertainty from sources such as occlusion, natural variations in lighting, and the pose of the animals.

We evaluate the verification task on a held out test set of 8576 pet images, from which all pairs were analyzed. On this set, point embeddings achieve 0.777 balanced accuracy. Binning the uncertainty measures and measuring correlation with the verification task (similarly as with N-digit MNIST), we find the correlation between performance and task accuracy to be 0.995. This experiment shows

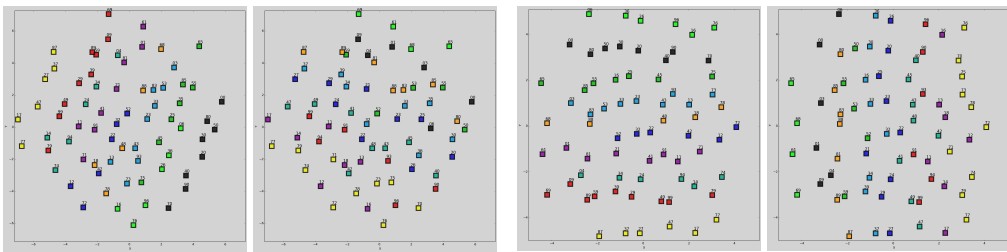

(a) Point embedding centroids.      (b) Hedged embedding centroids.

Figure 10: Uncertain embeddings self-organize. Class centroids for 2D point embeddings of 2-digit MNIST are shown, colored here by ones digit (left) and tens digit (right). The hedged instance embeddings have a structure where the classes self-organize in an axis aligned manner because of the diagonal covariance matrix used in the model.

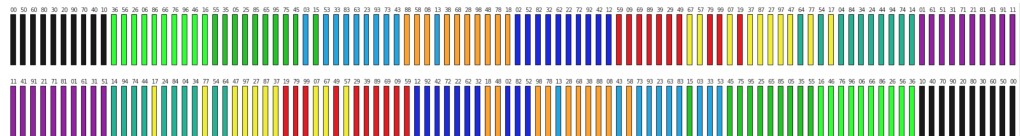

Figure 11: Embedding 2-digit MNIST onto the number line. **Top:** An ordering of class centroids produced with hedged instance embedding with Gaussian embeddings. **Bottom:** An ordering produced with point embedding. Centroids are colored according to least significant (ones) digit. The hedged instance embedding more often places classes that share attributes (in this case, a ones digit or tens digit).

that HIB scales to real-world problems with real-world sources of corruption, and provides evidence that HIB understands which embeddings are uncertain.

# D    ORGANIZATION OF THE LATENT EMBEDDING SPACE

As hedged instance embedding training progresses, it is advantageous for any subset of classes that may be confused with one another to be situated in the embedding space such that a given input image's embedding can strategically place probability mass. We observe this impacts the organization of the underlying space. For example, in the 2D embeddings shown in Figure 10, the class centers of mass for each class are roughly axis-aligned so that classes that share a tens' digit vary by x-coordinate, and classes that share a least significant (ones) digit vary by y-coordinate.

To further explore this idea, we embed 2-digit MNIST into a single dimension, to see how the classes get embedded along the number line. For hedged instance embedding, a single Gaussian embedding was chosen as the representation. We conjectured that because hedged instance embedding reduces objective loss by placing groups of confusing categories nearby one another, the resulting embedding space would be organized to encourage classes that share a tens or ones digit to be nearby. Figure 11 shows an example embedding learned by the two methods.

We assess the embedding space as follows. First the centroids for each of the 100 classes are derived from the test set embeddings. After sorting the classes, a count is made of adjacent class pairs that share a ones or tens digit, with the maximum possible being 99. The hedged embeddings outscored the point embeddings on each of the the four trials, with scores ranging from 76 to 80 versus scores of 42 to 74. Similarly, consider a *run* as a series of consecutive class pairs that share a ones or tens digit. The average run contains 4.6 classes with from hedged embeddings, and only 3.0 for point embeddings, as illustrated in Figure 11.

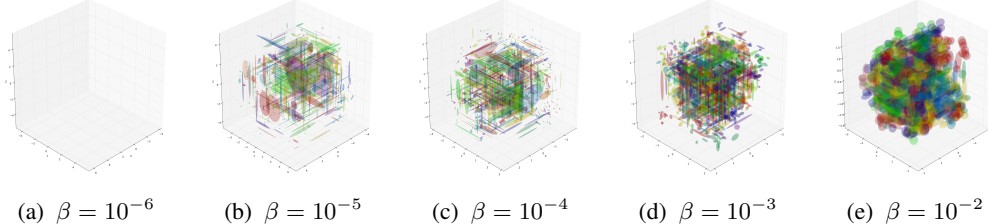

| (a) $\beta = 10^{-6}$ | (b) $\beta = 10^{-5}$ | (c) $\beta = 10^{-4}$ | (d) $\beta = 10^{-3}$ | (e) $\beta = 10^{-2}$ |
|---|---|---|---|---|

Figure 12: Impact of the weight on KL divergence. Each plot shows ellipsoids corresponding to hedged embeddings of corrupted test images from models mapping 3-digit MNIST to 3 dimensions. From left to right, the KL divergence weight increases by a factor of ten with each plot. When the weight $\beta$ is too small ($10^{-6}$), the embeddings approach points, and when too large ($10^{-2}$) embeddings approach the unit Gaussian.

|  | $N = 2, D = 2$ | | $N = 3, D = 3$ | |
|---|---|---|---|---|
| $\beta$ | 0 | $10^{-4}$ | 0 | $10^{-4}$ |
| **Verification AP** | | | | |
| clean | 0.986 | 0.988 | 0.993 | 0.992 |
| corrupt | 0.900 | 0.906 | 0.932 | 0.932 |
| **KNN Accuracy** | | | | |
| clean | 0.867 | 0.891 | 0.858 | 0.861 |
| corrupt | 0.729 | 0.781 | 0.685 | 0.730 |

|  | $N = 2, D = 2$ | | $N = 3, D = 3$ | |
|---|---|---|---|---|
| $\beta$ | 0 | $10^{-4}$ | 0 | $10^{-4}$ |
| **AP Correlation** | | | | |
| clean | 0.680 | 0.766 | 0.425 | 0.630 |
| corrupt | 0.864 | 0.836 | 0.677 | 0.764 |
| **KNN Correlation** | | | | |
| clean | 0.748 | 0.800 | -0.080 | 0.685 |
| corrupt | 0.636 | 0.609 | 0.183 | 0.549 |

| (a) Task performances. | (b) Uncertainty correlations. |
|---|---|

Table 5: Results for single Gaussian embeddings with and without KL divergence term. We report results for images with $N$ digits and using $D$ embedding dimensions.

## E    KL DIVERGENCE REGULARIZATION

The training objective (Equation 8) contains a regularization hyperparameter $\beta \geq 0$ controlling the weight of the KL divergence regularization that controls, from information theoretic perspective, bits of information encoded in the latent representation. In the main experiments, we consistently use $\beta = 10^{-4}$. In this Appendix, we explore the effect of this regularization by considering other values of $\beta$.

See Figure 12 for visualization of embeddings according to $\beta$. We observe that increasing KL term weight induces overall increase in variances of embeddings. For quantitative analysis on the impact of $\beta$ on main task performance and uncertainty quality, see Table 5 where we compare $\beta = 0$ and $\beta = 10^{-4}$ cases. We observe mild improvements in main task performances when the KL divergence regularization is used. For example, KNN accuracy improves from 0.685 to 0.730 by including the KL term for $N = 3$, $D = 3$ case. Improvement in uncertainty quality, measured in terms of Kendall's tau correlation, is more pronounced. For example, KNN accuracy and uncertainty are nearly uncorrelated when $\beta = 0$ under the $N = 3$, $D = 3$ setting (-0.080 for clean and 0.183 for corrupt inputs), while they are well-correlated when $\beta = 10^{-4}$ (0.685 for clean and 0.549 for corrupt inputs). KL divergence helps generalization, as seen by the main task performance boost, and improves the uncertainty measure by increasing overall variances of embeddings, as seen in Figure 12.

## F    DERIVATION OF VIB OBJECTIVE FOR STOCHASTIC EMBEDDING

Our goal is to train a discriminative model for match prediction on a pair of variables, $p(m|f(x_1), f(x_2))$, as opposed to predicting a class label, $p(y|x)$. Our VIB loss (Equation 8) follows straightforwardly from the original VIB, with two additional independence assumptions. In particular, we assume that the samples in the pair are independent, so $p(x_1, x_2) = p(x_1)p(x_2)$. We also assume the embeddings do not depend on the other input in the pair, $p(z_1, z_2|x_1, x_2) =$

$p(z_1|x_1)p(z_2|x_2)$. With these two assumptions, the VIB objective is given by the following:

$$I((z_1, z_2), m) - \beta I((z_1, z_2), (x_1, x_2)). \tag{10}$$

We variationally bound the first term using the approximation $q(m|z_1, z_2)$ of $p(m|z_1, z_2)$ as follows

$$I((z_1, z_2), m) = \int p(m, z_1, z_2) \log \frac{p(m, z_1, z_2)}{p(m)p(z_1, z_2)} \, \mathrm{d}m \, \mathrm{d}z_1 \, \mathrm{d}z_2 \tag{11}$$

$$= \int p(m, z_1, z_2) \log \frac{p(m|z_1, z_2)}{p(m)} \, \mathrm{d}m \, \mathrm{d}z_1 \, \mathrm{d}z_2 \tag{12}$$

$$= \int p(m, z_1, z_2) \log \frac{q(m|z_1, z_2)}{p(m)} \, \mathrm{d}m \, \mathrm{d}z_1 \, \mathrm{d}z_2 + \mathrm{KL}(p(m|z_1, z_2) \,\|\, q(m|z_1, z_2)) \tag{13}$$

$$\geq \int p(m, z_1, z_2) \log \frac{q(m|z_1, z_2)}{p(m)} \, \mathrm{d}m \, \mathrm{d}z_1 \, \mathrm{d}z_2 \tag{14}$$

$$= \int p(m, z_1, z_2) \log q(m|z_1, z_2) \, \mathrm{d}m \, \mathrm{d}z_1 \, \mathrm{d}z_2 + H(m) \tag{15}$$

$$\geq \int p(m, z_1, z_2) \log q(m|z_1, z_2) \, \mathrm{d}m \, \mathrm{d}z_1 \, \mathrm{d}z_2 \tag{16}$$

$$= \int p(m|x_1, x_2)p(z_1|x_1)p(z_2|x_2)p(x_1)p(x_2) \log q(m|z_1, z_2) \, \mathrm{d}m \, \mathrm{d}z_1 \, \mathrm{d}z_2 \, \mathrm{d}x_1 \, \mathrm{d}x_2. \tag{17}$$

The inequalities follow from the non-negativity of KL-divergence $\mathrm{KL}(\cdot)$ and entropy $H(\cdot)$. The final equality follows from our assumptions above.

The second term is variationally bounded using approximation $r(z_i)$ of $p(z_i|x_i)$ as follows:

$$I((z_1, z_2), (x_1, x_2)) = \int p(z_1, z_2, x_1, x_2) \log \frac{p(z_1, z_2|x_1, x_2)}{p(z_1, z_2)} \, \mathrm{d}z_1 \, \mathrm{d}z_2 \, \mathrm{d}x_1 \, \mathrm{d}x_2 \tag{18}$$

$$= \int p(z_1, z_2, x_1, x_2) \log \frac{p(z_1, z_2|x_1, x_2)}{r(z_1)r(z_2)} \, \mathrm{d}z_1 \, \mathrm{d}z_2 \, \mathrm{d}x_1 \, \mathrm{d}x_2$$
$$\quad - KL(p(z_1) \,\|\, r(z_1)) - KL(p(z_2) \,\|\, r(z_2)) \tag{19}$$

$$\leq \int p(z_1, z_2, x_1, x_2) \log \frac{p(z_1, z_2|x_1, x_2)}{r(z_1)r(z_2)} \, \mathrm{d}z_1 \, \mathrm{d}z_2 \, \mathrm{d}x_1 \, \mathrm{d}x_2 \tag{20}$$

$$= \int p(z_1, z_2|x_1, x_2)p(x_1, x_2) \log \frac{p(z_1, z_2|x_1, x_2)}{r(z_1)r(z_2)} \, \mathrm{d}z_1 \, \mathrm{d}z_2 \, \mathrm{d}x_1 \, \mathrm{d}x_2 \tag{21}$$

$$= \int p(z_1|x_1)p(x_1) \log \frac{p(z_1|x_1)}{r(z_1)} \, \mathrm{d}z_1 \, \mathrm{d}x_1$$
$$\quad + \int p(z_2|x_2)p(x_2) \log \frac{p(z_2|x_2)}{r(z_2)} \, \mathrm{d}z_2 \, \mathrm{d}x_2. \tag{22}$$

The inequality, again, follows from the non-negativity of KL-divergence, and the last equality follows from our additional independence assumptions.

Combining the two bounds, the VIB objective (Equation 10) for a fixed input pair $(x_1, x_2)$ is bounded from below by

$$\int p(z_1|x_1)p(z_2|x_2) \log q(m|z_1, z_2) \, \mathrm{d}z_1 \, \mathrm{d}z_2 \tag{23}$$

$$- \beta \left( \mathrm{KL}(p(z_1|x_1) \,\|\, r(z_1)) + \mathrm{KL}(p(z_2|x_2) \,\|\, r(z_2)) \right). \tag{24}$$

The negative of the above expression is identical to the loss $\mathcal{L}_{\mathrm{VIBEmb}}$ in Equation 8.

