# OpenReview forum: "Modeling Uncertainty with Hedged Instance Embeddings"
_ICLR.cc/2019/Conference_

### Official Review · AnonReviewer2 · 2018-10-22
**Review of "Modeling Uncertainty with Hedged Instance Embeddings"**

**Rating:** 7
**Confidence:** 5

**Review:**

While most works consider embedding as the problem of mapping an input into a point in an embedding space, paper 1341 considers the problem of mapping an input into a distribution in an embedding space. Computing the matching score of two inputs (e.g. two images) involves the following steps: (i) assuming a Gaussian distribution in the embedding space, computing the mean and standard deviation for each input, (ii) drawing a set of samples from each distribution, (3) computing the normalized distances between the samples and (iv) averaging to obtain a global score.

The proposed approach is validated on a new benchmark built on MNIST.

On the positive side:
-	The topic of injecting uncertainty in neural networks should be of broad interest to the ICLR community.
-	The paper is generally clear.
-	The qualitative evaluation provides intuitive results.

On the negative side:
-	The whole idea of drawing samples to compute the distance between two Gaussian distributions seems unnecessarily complicated. Why not computing directly a distance between distributions? There exist kernels between distributions, such as the Probability Product Kernel (PPK). See Jebara, Kondor, Howard “Probability product kernels”, JMLR’04. The PPK between two distributions p(x) and q(x) writes as: \int_x p^a(x) q^a(x) dx, where a is a parameter. When a=1, it is known as the Expected Likelihood Kernel (ELK). When a=1/2, this is known as the Hellinger or Bhattacharyya kernel (BK). In p and q are Gaussian distributions, then the PPK can be computed in closed form. If p and q are mixtures of Gaussians, then the ELK can be computed in closed form.
-	The Mixture of Gaussians embedding extension is lacking in details. How does the network generate C Gaussian distributions? By having 2C output branches generating C means and C standard deviation vectors?
-	It might be useful to provide more details about why the self-similarity measure makes sense as an uncertainty measure. In its current state, the paper does not provide much intuition and it took me some time to understand (I actually understood when I made the connection with the ELK). Also, why not using a simpler measure of uncertainty such as the trace of the covariance matrix?
-	The experiments are lacking in some respects:
o	It would be useful to report results without the VIB regularization.
o	The focus on the cases D=2 and D=3 (embedding in a 2D or 3D space) shades some doubt on the practical usefulness of this framework in a higher-dimensional case.

Miscellaneous:
-	It seems there is a typo between equations (4) and (5). It should write z_1^{(k_1)} \sim p(z_1|x_1)

---

In their rebuttal, the authors satisfyingly addressed my concerns. Hence, I am upgrading my overall rating.

---

> ### Author Response · Authors · 2018-11-12
> **Author Response 1/2**
>
> We thank the reviewers for recognising the importance of the problem (R2) and finding the paper well-written (R2, R3). We have revised the submission according to reviewers’ suggestions and proposals (see summary of updates below). We respond to each reviewer’s comments below.
>
>
> = Summary of updates in revision:
>
> - Discussion of conceptual inapplicability of MC dropout for probabilistic embedding in Section 3 (R1).
> - Discussion of the intuition behind self-similarity for uncertainty measure and its conceptual advantage over the trace of covariance matrix (R2).
> - Description of network architecture for MoG embedding in Section 2.2 (R2).
> - New appendix section E for qualitative and quantitative analysis of the impact of KL divergence regularization term (R2).
> - Added (N=2, D=3) columns in tables 1 and 2.
> - Typos (R2,3).
>
>
> = R1: Comparison against “existing uncertainty methods like dropout”.
>
> Randomness in MC dropout is independent of input. It is designed to measure model uncertainty (epistemic uncertainty). On the other hand, our model is designed to measure input uncertainty (aleatoric uncertainty). They are conceptually distinct methods.  We have added this discussion in Section 3, “Probabilistic DNNs” paragraph of the revision.
>
>
> = R1: Unlike MC dropout, “hyperparameters [such as number of components for MoG] are a pain point”.
>
> We have found results to be fairly insensitive to number of mixture components in the MoG. Note that MC dropout also has parameters to tune!
>
>
> = R1: How should I choose the number of components given that there are ten possibilities for each digit?
>
> Do cross-validation if performance is critical. We have shown, however, that both a single Gaussian and two-component MoG perform well in our setup (Section 4.3).
>
>
> = R2: Sampling based similarity computation is “complicated”. Why not compute analytic distances for Gaussians like Expected Likelihood Kernel or Hellinger Kernel?
>
> Our similarity computation based on distance samples is motivated by the contrastive loss metric learning objective (Section 2.1), where Euclidean distances between embeddings encode similarity of inputs. Compared to divergence (KL or JS), inner product (ELK), or Hellinger kernel based distances, HIB is designed to more directly represent distributions of Euclidean distances on the embedding space (through the match probability, Equation 2). However, we agree that it would be interesting to explore no-sampling alternatives in future work.
>
>
> = R2: What is the intuition behind self-similarity for uncertainty? Why not use the trace of the covariance matrix for uncertainty?
>
> The self-similarity uncertainty measure starts from the intuition that for ambiguous inputs, their embeddings will span diverse semantic classes (as in Figure 1b). To quantify this, we have defined self-similarity as the chance that given an input x and two independent samples (z1, z2) from its embedding p(z|x), they belong to the same semantic cluster (i.e., their match probability).
>  We do not use volumetric uncertainty measures like trace or determinant of covariance matrix because it does not make sense for multi-modal distributions like MoG. We have updated Section 2.4 with this discussion.

---

> > ### Author Response · Authors · 2018-11-12
> > **Author Response 2/2**
> >
> >
> > = R2: Is MoG implemented “by having 2C output branches generating C means and C standard deviation vectors”?
> >
> > Yes, your description is correct. We have added the detail in Section 2.2, “MoG embedding” paragraph in the revision.
> >
> >
> > = R2: “It would be useful to report results without the VIB regularization.”
> >
> > We have added the results in the new appendix section E, “KL Divergence Regularization”. We confirm that the KL term improves generalisation for the main tasks (verification and recognition) and better calibrates the uncertainty measure.
> >
> >
> > = R2: Doubt “practical usefulness” in a higher-dimensional case than D=2 or 3 in the paper.
> >
> > The space and time complexity of increasing D scale only linearly with D. We focus on D=2 and 3 because these compact embeddings stress the network's ability to discriminate well across many (100 or 1000) classes. We have successfully trained HIB with larger dimensional embeddings (D=6) and in that case HIB also exhibits good correlations between uncertainty and task performance. However, the task accuracy with N-digit-MNIST begins to saturate, making it difficult to further explore the relationship between the uncertainty measure and task performance.
> >
> >
> > = R3: Evaluation should confirm that the “uncertainty measure actually affects the downstream task in a known manner” for example by showing it helps certain “active learning framework”.
> >
> > While we agree it would be interesting to see if collecting additional views of an ambiguous input reduces its uncertainty, this is not always an option in practice (e.g., face recognition from a single image). Therefore, we focus on the task of assessing when a model should say "I don't know" given a single input.
> >
> >
> > = R3: In Figure 5, correlation between the embedding uncertainty and KNN will be high regardless of the quality of embedding.
> >
> > The aim of Figure 5 is not to measure the quality of embedding (which is measured in Section 4.3); it is to measure the quality of our uncertainty (i.e., higher uncertainty for low-performance inputs). Figure 5 is confirming that our measure of uncertainty is indeed a good predictor of downstream task performances. If this misses the point, please further clarify your concern.

---

### Official Review · AnonReviewer3 · 2018-10-30
**Great paper! Could use more uncertainty-measuring application / experiments**

**Rating:** 7
**Confidence:** 3

**Review:**

pros: The  paper is well-written and well-motivated. It seems like uncertain-embeddings will be a valuable tool as we continue to extend deep learning to Bayesian applications, and the model proposed here seems to work well, qualitatively. Additionally the paper is well-written, in that every step used to construct the loss function and training seem well motivated and generally intuitive, and the simplistic CNN and evaluations give confidence that this is not a random result.

cons: I think the quantitative results are not as impressive as I would have expected, and I think it is because the wrong thing is being evaluated. It would make the results more  impressive to try to use these embeddings in some active learning framework, to see if proper understanding of uncertainty helps in a task where a good uncertainty measure actually affects the downstream task in a known manner. Additionally, I don't think Fig 5 makes sense, since you are using the embeddings for the KNN task, then measuring correlation between the embedding uncertainty and KNN, which might be a high correlation without the embedding being good.

Minor comments:
 - Typo above (5) on page 3.
 - Appendix line under (12), I think dz1 and dz2 should be after the KL terms.

Reviewer uncertainty: I am not familiar enough with the recent literature on this topic to judge novelty.

---

> ### Author Response · Authors · 2018-11-12
> **Author Response 1/2**
>
> We thank the reviewers for recognising the importance of the problem (R2) and finding the paper well-written (R2, R3). We have revised the submission according to reviewers’ suggestions and proposals (see summary of updates below). We respond to each reviewer’s comments below.
>
>
> = Summary of updates in revision:
>
> - Discussion of conceptual inapplicability of MC dropout for probabilistic embedding in Section 3 (R1).
> - Discussion of the intuition behind self-similarity for uncertainty measure and its conceptual advantage over the trace of covariance matrix (R2).
> - Description of network architecture for MoG embedding in Section 2.2 (R2).
> - New appendix section E for qualitative and quantitative analysis of the impact of KL divergence regularization term (R2).
> - Added (N=2, D=3) columns in tables 1 and 2.
> - Typos (R2,3).
>
>
> = R1: Comparison against “existing uncertainty methods like dropout”.
>
> Randomness in MC dropout is independent of input. It is designed to measure model uncertainty (epistemic uncertainty). On the other hand, our model is designed to measure input uncertainty (aleatoric uncertainty). They are conceptually distinct methods.  We have added this discussion in Section 3, “Probabilistic DNNs” paragraph of the revision.
>
>
> = R1: Unlike MC dropout, “hyperparameters [such as number of components for MoG] are a pain point”.
>
> We have found results to be fairly insensitive to number of mixture components in the MoG. Note that MC dropout also has parameters to tune!
>
>
> = R1: How should I choose the number of components given that there are ten possibilities for each digit?
>
> Do cross-validation if performance is critical. We have shown, however, that both a single Gaussian and two-component MoG perform well in our setup (Section 4.3).
>
>
> = R2: Sampling based similarity computation is “complicated”. Why not compute analytic distances for Gaussians like Expected Likelihood Kernel or Hellinger Kernel?
>
> Our similarity computation based on distance samples is motivated by the contrastive loss metric learning objective (Section 2.1), where Euclidean distances between embeddings encode similarity of inputs. Compared to divergence (KL or JS), inner product (ELK), or Hellinger kernel based distances, HIB is designed to more directly represent distributions of Euclidean distances on the embedding space (through the match probability, Equation 2). However, we agree that it would be interesting to explore no-sampling alternatives in future work.
>
>
> = R2: What is the intuition behind self-similarity for uncertainty? Why not use the trace of the covariance matrix for uncertainty?
>
> The self-similarity uncertainty measure starts from the intuition that for ambiguous inputs, their embeddings will span diverse semantic classes (as in Figure 1b). To quantify this, we have defined self-similarity as the chance that given an input x and two independent samples (z1, z2) from its embedding p(z|x), they belong to the same semantic cluster (i.e., their match probability).
>  We do not use volumetric uncertainty measures like trace or determinant of covariance matrix because it does not make sense for multi-modal distributions like MoG. We have updated Section 2.4 with this discussion.

---

> > ### Author Response · Authors · 2018-11-12
> > **Author Response 2/2**
> >
> >
> > = R2: Is MoG implemented “by having 2C output branches generating C means and C standard deviation vectors”?
> >
> > Yes, your description is correct. We have added the detail in Section 2.2, “MoG embedding” paragraph in the revision.
> >
> >
> > = R2: “It would be useful to report results without the VIB regularization.”
> >
> > We have added the results in the new appendix section E, “KL Divergence Regularization”. We confirm that the KL term improves generalisation for the main tasks (verification and recognition) and better calibrates the uncertainty measure.
> >
> >
> > = R2: Doubt “practical usefulness” in a higher-dimensional case than D=2 or 3 in the paper.
> >
> > The space and time complexity of increasing D scale only linearly with D. We focus on D=2 and 3 because these compact embeddings stress the network's ability to discriminate well across many (100 or 1000) classes. We have successfully trained HIB with larger dimensional embeddings (D=6) and in that case HIB also exhibits good correlations between uncertainty and task performance. However, the task accuracy with N-digit-MNIST begins to saturate, making it difficult to further explore the relationship between the uncertainty measure and task performance.
> >
> >
> > = R3: Evaluation should confirm that the “uncertainty measure actually affects the downstream task in a known manner” for example by showing it helps certain “active learning framework”.
> >
> > While we agree it would be interesting to see if collecting additional views of an ambiguous input reduces its uncertainty, this is not always an option in practice (e.g., face recognition from a single image). Therefore, we focus on the task of assessing when a model should say "I don't know" given a single input.
> >
> >
> > = R3: In Figure 5, correlation between the embedding uncertainty and KNN will be high regardless of the quality of embedding.
> >
> > The aim of Figure 5 is not to measure the quality of embedding (which is measured in Section 4.3); it is to measure the quality of our uncertainty (i.e., higher uncertainty for low-performance inputs). Figure 5 is confirming that our measure of uncertainty is indeed a good predictor of downstream task performances. If this misses the point, please further clarify your concern.

---

> > > ### Comment · AnonReviewer3 · 2018-11-22
> > > **Response to rebuttal**
> > >
> > > = While we agree it would be interesting to see if collecting additional views of an ambiguous input reduces its uncertainty, this is not always an option in practice (e.g., face recognition from a single image). Therefore, we focus on the task of assessing when a model should say "I don't know" given a single input.
> > >
> > > Well, you can imagine a reinforcement learning kind of scenario, or bandit scenario, where you are accomplishing some task where multiple samples increases certainty, but you have a budget on how many samples you can take. So, using your measure of uncertainty, you can then come up with an "optimal sampling schedule", which could really improve the performance of such a task.
> > >
> > >
> > > =The aim of Figure 5 is not to measure the quality of embedding (which is measured in Section 4.3); it is to measure the quality of our uncertainty (i.e., higher uncertainty for low-performance inputs). Figure 5 is confirming that our measure of uncertainty is indeed a good predictor of downstream task performances. If this misses the point, please further clarify your concern.
> > >
> > > I still think such a thing should be measured against a task that did not use the embedding; you can still compare uncertainty and task performance in the same way. It is a bit nitpicky, but I think it would make the evaluation results more meaningful.

---

### Official Review · AnonReviewer1 · 2018-11-01
**Modelling Uncertainty with Hedged Instance Embeddings**

**Rating:** 7
**Confidence:** 3

**Review:**

# Summary
Paper proposes an alternative to current point embedding and a technique to train them. Point embedding are conventional embedding where an input x is deterministically mapped to a vector in embedding space.

i.e         f(x) = z where f may be a parametric function or trained Neural network.

Note that this point embedding means that every x is assigned a unique z, this might be an issue in cases where x is confusing for example if x is an image in computer vision pipeline then x may be occluded etc. In such cases paper argues that assigning a single point as embedding is not a great option.

Paper says that instead of assigning a single point it's better to assign smear of points (collection of points coming from some distributions like Gaussian and mixture of Gaussian etc)

They provide a technique based on variational inference to train the network to produce such embeddings. They also propose a new dataset made out of MNIST to test this concept.

# Concerns

Although they have results to back up their claim on their proposed dataset and problem. They have not compared with many existing uncertainty methods like dropout. (But I’m not sure if such a comparison is relevant here)
Unlike Kendall method or dropout method, hyperparameters here are a pain point for me, i.e how many Gaussians should I consider in my mixture of Gaussian to create the embeddings (results will depend upon that)
I.e consider the following scenario
The first digit is occluded and can be anything 1,2,3,4,5,6,7,8,9,0 should I use only one Gaussian to create my embeddings like they have shown in the paper for this example, or should I choose 10 gaussian each centered about one of the digits, which might help in boosting the performance?

---

> ### Author Response · Authors · 2018-11-12
> **Author Response 1/2**
>
> We thank the reviewers for recognising the importance of the problem (R2) and finding the paper well-written (R2, R3). We have revised the submission according to reviewers’ suggestions and proposals (see summary of updates below). We respond to each reviewer’s comments below.
>
>
> = Summary of updates in revision:
>
> - Discussion of conceptual inapplicability of MC dropout for probabilistic embedding in Section 3 (R1).
> - Discussion of the intuition behind self-similarity for uncertainty measure and its conceptual advantage over the trace of covariance matrix (R2).
> - Description of network architecture for MoG embedding in Section 2.2 (R2).
> - New appendix section E for qualitative and quantitative analysis of the impact of KL divergence regularization term (R2).
> - Added (N=2, D=3) columns in tables 1 and 2.
> - Typos (R2,3).
>
>
> = R1: Comparison against “existing uncertainty methods like dropout”.
>
> Randomness in MC dropout is independent of input. It is designed to measure model uncertainty (epistemic uncertainty). On the other hand, our model is designed to measure input uncertainty (aleatoric uncertainty). They are conceptually distinct methods.  We have added this discussion in Section 3, “Probabilistic DNNs” paragraph of the revision.
>
>
> = R1: Unlike MC dropout, “hyperparameters [such as number of components for MoG] are a pain point”.
>
> We have found results to be fairly insensitive to number of mixture components in the MoG. Note that MC dropout also has parameters to tune!
>
>
> = R1: How should I choose the number of components given that there are ten possibilities for each digit?
>
> Do cross-validation if performance is critical. We have shown, however, that both a single Gaussian and two-component MoG perform well in our setup (Section 4.3).
>
>
> = R2: Sampling based similarity computation is “complicated”. Why not compute analytic distances for Gaussians like Expected Likelihood Kernel or Hellinger Kernel?
>
> Our similarity computation based on distance samples is motivated by the contrastive loss metric learning objective (Section 2.1), where Euclidean distances between embeddings encode similarity of inputs. Compared to divergence (KL or JS), inner product (ELK), or Hellinger kernel based distances, HIB is designed to more directly represent distributions of Euclidean distances on the embedding space (through the match probability, Equation 2). However, we agree that it would be interesting to explore no-sampling alternatives in future work.
>
>
> = R2: What is the intuition behind self-similarity for uncertainty? Why not use the trace of the covariance matrix for uncertainty?
>
> The self-similarity uncertainty measure starts from the intuition that for ambiguous inputs, their embeddings will span diverse semantic classes (as in Figure 1b). To quantify this, we have defined self-similarity as the chance that given an input x and two independent samples (z1, z2) from its embedding p(z|x), they belong to the same semantic cluster (i.e., their match probability).
>  We do not use volumetric uncertainty measures like trace or determinant of covariance matrix because it does not make sense for multi-modal distributions like MoG. We have updated Section 2.4 with this discussion.

---

> > ### Author Response · Authors · 2018-11-12
> > **Author Response 2/2**
> >
> >
> > = R2: Is MoG implemented “by having 2C output branches generating C means and C standard deviation vectors”?
> >
> > Yes, your description is correct. We have added the detail in Section 2.2, “MoG embedding” paragraph in the revision.
> >
> >
> > = R2: “It would be useful to report results without the VIB regularization.”
> >
> > We have added the results in the new appendix section E, “KL Divergence Regularization”. We confirm that the KL term improves generalisation for the main tasks (verification and recognition) and better calibrates the uncertainty measure.
> >
> >
> > = R2: Doubt “practical usefulness” in a higher-dimensional case than D=2 or 3 in the paper.
> >
> > The space and time complexity of increasing D scale only linearly with D. We focus on D=2 and 3 because these compact embeddings stress the network's ability to discriminate well across many (100 or 1000) classes. We have successfully trained HIB with larger dimensional embeddings (D=6) and in that case HIB also exhibits good correlations between uncertainty and task performance. However, the task accuracy with N-digit-MNIST begins to saturate, making it difficult to further explore the relationship between the uncertainty measure and task performance.
> >
> >
> > = R3: Evaluation should confirm that the “uncertainty measure actually affects the downstream task in a known manner” for example by showing it helps certain “active learning framework”.
> >
> > While we agree it would be interesting to see if collecting additional views of an ambiguous input reduces its uncertainty, this is not always an option in practice (e.g., face recognition from a single image). Therefore, we focus on the task of assessing when a model should say "I don't know" given a single input.
> >
> >
> > = R3: In Figure 5, correlation between the embedding uncertainty and KNN will be high regardless of the quality of embedding.
> >
> > The aim of Figure 5 is not to measure the quality of embedding (which is measured in Section 4.3); it is to measure the quality of our uncertainty (i.e., higher uncertainty for low-performance inputs). Figure 5 is confirming that our measure of uncertainty is indeed a good predictor of downstream task performances. If this misses the point, please further clarify your concern.

---

### Meta-Review · Area_Chair1 · 2018-12-06
**Point embeddings changed to distribution embeddings to model uncertainty.**

**Confidence:** 2
**Recommendation:** Accept (Poster)

**Metareview:**

This work presents a method to model embeddings as distributions, instead of points, to better quantify uncertainty. Evaluations are carried out on a new dataset created from mixtures of MNIST digits, including noise (certain probability of occlusions), that introduce ambiguity, using a small "toy" neural network that is incapable of perfectly fitting the data, because authors mention that performance difference lessens when the network is complex enough to almost perfectly fit the data.

Reviewer assessment is unanimously accept, with the following points:

Pros:
+ "The topic of injecting uncertainty in neural networks should be of broad interest to the ICLR community."
+ "The paper is generally clear."
+ "The qualitative evaluation provides intuitive results."

Cons:
- Requirement of drawing samples may add complexity. Authors reply that alternatives should be studied in future work.
- No comparison to other uncertainty methods, such as dropout. Authors reply that dropout represents model uncertainty and not data uncertainty, but do not carry out an experiment to compare (i.e. sample from model leaving dropout activated during evaluation).
- No evaluation in larger scale/dimensionality datasets. Authors mention method scales linearly, but how practical or effective this method is to use on, say, face recognition datasets, is unclear.

As the general reviewer consensus is accept, Area Chair is recommending Accept; However, Area Chair has strong reservations because the method is evaluated on a very limited dataset, with a toy model designed to exaggerate differences between techniques. Essentially, the toy evaluation was designed to get the results the authors were looking for. A more thorough investigation would use more realistic sized network models on true datasets.